# Development of an Artificial-Intelligence-Based Tool for Automated Assessment of Cellularity in Bone Marrow Biopsies in Ph-Negative Myeloproliferative Neoplasms

**DOI:** 10.3390/cancers16091687

**Published:** 2024-04-26

**Authors:** Giuseppe D’Abbronzo, Antonio D’Antonio, Annarosaria De Chiara, Luigi Panico, Lucianna Sparano, Anna Diluvio, Antonello Sica, Gino Svanera, Renato Franco, Andrea Ronchi

**Affiliations:** 1Department of Mental and Physical Health and Preventive Medicine, Università degli Studi della Campania “Luigi Vanvitelli”, 80138 Naples, Italy; dabbronzogiuseppe@gmail.com (G.D.); anna.diluvio@gmail.com (A.D.); andrea.ronchi@unicampania.it (A.R.); 2Pathology Unit, Hospital “Ospedale del Mare”, 80147 Naples, Italy; ada66@inwind.it; 3Histopathology of lymphomas and Sarcoma SSD, Istituto Nazionale dei Tumori I.R.C.C.S. Fondazione “Pascale”, 80131 Naples, Italy; a.dechiara@istitutotumori.na.it; 4Pathology Unit, Hospital “Monaldi”, 80131 Naples, Italy; luigi.panico@ospedaledeicolli.it; 5Pathology Unit, Hospital “Andrea Tortora”, 82100 Pagani, Italy; luciannasparano@gmail.com; 6Haematology and Oncology Unit, Vanvitelli Hospital, 80131 Naples, Italy; antonello.sica@fastwebnet.it; 7Haematology Unit, ASL Na2 North, 80014 Giugliano, Italy; gsvanemato@inwind.it; 8Pathology Unit, Vanvitelli Hospital, 80138 Naples, Italy

**Keywords:** digital pathology, artificial intelligence, cellularity, myeloproliferative neoplasms, bone marrow biopsy

## Abstract

**Simple Summary:**

In recent years, artificial intelligence has been used increasingly in medical practice. Technological progress has made it possible to digitize the slides of histological preparations, allowing the use of image processing and AI technologies in surgical pathology. This potentially reduces variability and improves the uniformity of some histological evaluations, such as cellularity assessment in bone marrow biopsies, which is traditionally performed visually by expert human observers, resulting in inter-observer and intra-observer variability. In this work, we developed an accurate AI-based tool for the automated quantification of cellularity in BMB histology and compared its performances with cellularity estimates from five expert hematopathologists. Our results showed the robustness of our model across users and two different scanners for digitized image generation.

**Abstract:**

The cellularity assessment in bone marrow biopsies (BMBs) for the diagnosis of Philadelphia chromosome (Ph)-negative myeloproliferative neoplasms (MPNs) is a key diagnostic feature and is usually performed by the human eyes through an optical microscope with consequent inter-observer and intra-observer variability. Thus, the use of an automated tool may reduce variability, improving the uniformity of the evaluation. The aim of this work is to develop an accurate AI-based tool for the automated quantification of cellularity in BMB histology. A total of 55 BMB histological slides, diagnosed as Ph- MPN between January 2018 and June 2023 from the archives of the Pathology Unit of University “Luigi Vanvitelli” in Naples (Italy), were scanned on Ventana DP200 or Epredia P1000 and exported as whole-slide images (WSIs). Fifteen BMBs were randomly selected to obtain a training set of AI-based tools. An expert pathologist and a trained resident performed annotations of hematopoietic tissue and adipose tissue, and annotations were exported as .tiff images and .png labels with two colors (black for hematopoietic tissue and yellow for adipose tissue). Subsequently, we developed a semantic segmentation model for hematopoietic tissue and adipose tissue. The remaining 40 BMBs were used for model verification. The performance of our model was compared with an evaluation of the cellularity of five expert hematopathologists and three trainees; we obtained an optimal concordance between our model and the expert pathologists’ evaluation, with poorer concordance for trainees. There were no significant differences in cellularity assessments between two different scanners.

## 1. Introduction

Artificial intelligence (AI) is increasingly being used in clinical practice and translational medical research. Various AI systems have been employed to improve the diagnosis and treatment of diseases such as diabetes [1] and cancer [2]. In surgical pathology, AI technologies have been facilitated by the ability to digitize slides. The first guideline for the production of whole-slide image (WSI) systems for clinical practice [3] was developed in 2016 by the Food and Drug Administration (FDA), who set out technical standards that ensured the production of acceptable images (varying color intensity, different cutting thickness) even in the presence of uneven tissue or tissue from different laboratories. Additionally, the WSI quality obtained from an approved scanner was required to have appropriate focus quality, color reproducibility, and spatial resolution to accurately reflect the quality of the original histology slides. Several commercial WSI and digital pathology systems have been approved by the FDA for diagnostic use and have received CE marks for in vitro diagnosis [4]. Digitized slides that meet these standards can be used to evaluate specific histological features such as the number of mitoses [5], biomarkers in breast tumor settings [6], or histological activity in ulcerative colitis [7]. Recently, an AI-supported diagnostic tool for prostate cancer based on digitized pathology slides (Paige Prostate) was approved by the FDA [8]. Despite these regulatory processes, variability in scanner performance has been shown to influence AI-based assessments [9,10].

Myeloproliferative neoplasms (MPNs) are a heterogeneous group of bone marrow disorders characterized by the abnormal proliferation of one or more myeloid cell lines, and include conditions like polycythemia vera (PV), essential thrombocythemia (ET), and primary myelofibrosis (PM) [11]. These diseases, collectively referred to as Philadelphia chromosome-negative (Ph-) MPNs, often require a complex diagnostic process involving clinical, laboratory, genetic, and histological data. In particular, the assessment of cellularity in bone marrow biopsies (BMBs) is a key diagnostic feature in MPNs [11]. However, the manual nature of the assessments, typically conducted through an optical microscope, results in significant inter-observer and intra-observer discrepancies [12,13,14,15]. Human-based cellularity assessments are further confounded by the presence of inflammation, necrosis, fibrosis, and specimen artifacts [16]. Consequently, concordance for MPN diagnoses, even between expert pathologists, ranges from 53%, based only on morphology, to 88% when paired with clinical and mutational data [17], and is poorly reproducible and dependent on pathologist experience [18,19]. Automated assessments of cellularity, like other computational pathology tools [20], would be advantageous not only due to providing objective assessments, but also due to improving laboratory workflow, reducing pathologist assessment time, and enhancing the training and skill development of young pathologists [21,22]. 

Prior work has explored automated assessment of BMBs in heterogenous patient groups. Nielsen et al. [23] developed a segmentation system for BMB components, classifying them as yellow bone marrow (YBM) and red bone marrow (RBM), and then calculating cellularity in specific areas of interest in BMBs from eight subjects. Van Eekelen et al. [24] developed a segmentation system for different cell lines present in the bone marrow (erythropoiesis, myelopoiesis, megakaryocytes, lipocytes, trabecular bone, and erythrocytes), estimating the variation in cellularity in BMBs with age. Both studies achieved an optimal degree of agreement with a human-based evaluation of cellularity, proving that hematopoietic tissue can be quickly and easily distinguished from adipose tissue and digital image processing can provide an accurate and objective measurement of the examined region. However, they did not address the utility of such tools in a clinical scenario across multiple users with different levels of expertise or across different scanners for digitized image generation. The aim of this work was to develop and train an AI-based tool for the automated quantification of cellularity in patients with Ph- MPNs, validate it against multiple expert histopathologist assessments, and use it to evaluate the performance of pathology trainees. In addition, model usability was appraised by users and model performance was compared between digitized slides from two different scanners.

## 2. Materials and Methods

### 2.1. Cases Selection and Digital Images Scanning

We obtained bone marrow biopsy (BMB) histological slides diagnosed as Ph-MPN between January 2018 and June 2023 from the archives of the Pathology Unit of University “Luigi Vanvitelli” in Naples (Italy). The inclusion criteria were the following: age of the patient ≥ 18 years; biopsy frustule length ≥ 1 cm; availability of clinical data (age and histological and clinical diagnosis), and histological diagnosis of Ph-MPN. We chose BMBs with diagnoses of Ph-MPN for their significant variations in bone marrow cellularity. Conditions such as PV and pre-fibrotic PM typically present with increased cellularity, whereas other forms, such as fibrotic PM, can exhibit reduced cellularity [11].

A total of 86 BMBs were initially retrieved from archives and reviewed by an experienced hematopathologist (AR). Of these, 55 cases met the inclusion criteria and were included in the study. A histological slide stained with hematoxylin and eosin was scanned for each case with a Ventana DP200 Slide Scanner (Roche Diagnostics S.p.A., Monza, Italy) and exported as a whole-slide image (WSI). Subsequently, scanned slides were randomly divided into a training set, including 15 BMBs, and a validation set, including 40 BMBs.

### 2.2. Production of the Training Set

An expert pathologist (AR), assisted by a trained resident (GDA), annotated hematopoietic and adipose tissues on the 15 BMBs selected for the training set using QuPath v 4.3 software [25]. Then, annotations were exported as .tiff images (resolution 256 × 256, 20× magnification) and .png labels (Figure 1) with three main colors (black for hematopoietic tissue; yellow for adipose tissue; and white for regions that must be ignored during the training, such as bone tissue, red blood cells, artifacts, etc.). A specific script for Qupath [26] was used to export the annotation, providing 2617 images and 2617 labels that constituted our training set. 

A semantic segmentation model for hematopoietic tissue and adipose tissue was developed using a fully convolutional network (FCN) based on InceptionV3 architecture, utilized up to the “mixed7” layer. This architecture was enhanced with a series of convolutional and deconvolutional layers, concluding with a 1 × 1 convolution for pixel-by-pixel classification using softmax activation. The model featured a total of nine main custom layers post-inception and was trained over 25 epochs using the Adam optimizer with categorical cross-entropy as the loss function.

### 2.3. Model Testing

Firstly, the cellularity of the 40 BMBs selected for the validation set was evaluated by 5 expert pathologists (RF, ADA, ADC, LP, LS) and 3 young pathologists (Resident1 Resident2, Resident3) using an optical microscope, as is routinely performed in clinical practice. Subsequently, two scripts for model application and tissue segmentation were developed. The first script was developed using the FAST library for Python [27] and segmented the WSI image into 256 × 256 patches at 20× magnification. It applied the model to each patch, identifying adipose and hematopoietic tissues; extracted the corresponding labels; and reconstructed the slide with the segmentation results overlaid on the tissue (see Figure 2). The second script calculated the cellularity as the ratio of the area segmented as hematopoietic tissue to the total area of both hematopoietic and adipose tissues.
$$Cellularity=\frac{Segmented\;area\;of\;hematopoietic\;tissue}{Segmented\;area\;of\;hematopoietic\;tissue+Segmented\;area\;of\;adipose\;tissue}\times 100\%$$

The agreement between the hematopathologists and the model, as well as between the residents and the model, was evaluated using Lin’s concordance correlation coefficient. Furthermore, agreement between the hematopathologist and the model was also evaluated using the Bland–Altman plot.

Six months later, the expert pathologist who annotated the training images (A.R.-user1), one of the expert pathologists who was compared with the model (R.F.-user2), and a resident (Resident2-user3) were trained to use the proposed model on a set of 20 cases from the BMB series. Subsequently, we asked these three users to rate the model on a scale from 1 to 5 (1: insufficient–5: optimal) in terms of usability (learning difficulties and autonomy in using the model), rapidity, and accuracy of evaluations. Finally, in order to test the performance of our model with different scanners, we randomly selected 5 BMBs from the validation set and rescanned them using a different scanner (Epredia P1000) (Epredia, Portsmouth, NH, USA). We then compared the performance of our model with the WSIs obtained from both the Ventana DP200 and the Epredia P1000 in terms of segmentation (assessed visually) and cellularity.

### 2.4. Statistical Analysis

To verify the effectiveness of our tool, we compared the performance of our model with the cellularity assessment performed by 5 pathologists (R.F.; A. D’antonio; A. De Chiara, L.S., and L.P.) with at least ten years of experience in the field of hematopathology. We asked them to express a single percentage value of cellularity in each BMB. Firstly, we measured the agreement between hematopathologists in pairs in order to test for interobserver variability in the assessment of cellularity; then, we measured the agreement between our model and each pathologist.

Furthermore, in order to show the poor reproducibility of evaluations of cellularity in BMBs, we asked medical residents (Resident 1, Resident 2, and Resident 3) to express their cellularity assessments on the same cases submitted to the expert pathologists, and we compared their evaluation with the performance of our model.

To measure agreement between the hematopathologists, between the hematopathologists and the model, and between the residents and the model, we chose to use Lin’s concordance correlation coefficient (CCC) [28], which considers both the correlation of the assessments (Pearson correlation) and the average and variance of the measurements: $$CCC:\;\frac{2p\sigma_{x}\sigma_{y}}{\sigma_{x}^{2}+\sigma_{y}^{2}+{\left(\mu_{x}+\mu_{y}\right)}^{2}}$$


*p: Pearson correlation coefficient*




$\sigma_{x}^{2};\sigma_{y}^{2}$

*: variance of measurement*




$\mu_{x};\mu_{y}$

*: average of measurement*


In this way, the agreement between pathologists’ model measurements was evaluated in terms of the evaluated percentages, penalizing both systematic and random differences.

According to Altman’s definition [29], we considered optimal agreement to be characterized by a CCC greater than 0.8 and substantial agreement by a CCC between 0.6 and 0.8. Subsequently, in order to further evaluate the agreement between our model and the pathologists, we performed another statistical test using Bland–Altman plots.

Bland–Altman plots compare the averages of measurements, provide graphical representations of the agreement between the system of measurement, and identify any systematic trends [30]. In detail, agreement ranges were determined by calculating the averages and standard deviations of differences between measurements provided by the AI system and the pathologist. Then, we calculated the upper limit of agreement (ULA) and the lower agreement limit (LAL) according to the following formulas: $$ULA=\bar{d}+1.96\times S_{d}$$
$$LAL=\bar{d}-1.96\times S_{d}$$

$\bar{d}:$ average between measurement difference;

$S_{d}:$ standard deviation of measurement difference.

The calculation was based on the assumption of a normal distribution of the differences between measurements, covering 95% of the expected differences between the measurements of the model and the measurements of the pathologists. 

### 2.5. Hardware and Software

The annotations and classifications of the training set were carried out using QuPath software, version 4.3; the images and masks were exported using a specific script in Groovy language. All segmentation models were developed using Python 3.10.12 on the Google Colaboratory platform with a pro profile for the use of additional RAM and GPUs provided by the platform (V100). The scripts for model verification, cellularity calculation, and statistical analysis were developed in Python 3.10.12 using the Anaconda^®^ distribution, with Jupyter Notebook as the programming environment. All computer processing procedures were performed on an Alienware Aurora^®^ R11 hardware (Alienware, Dell Inc., Round Rock, TX, USA) with an Intel(R) Core (TM) i5-10600KF CPU at 4.10 GHz, 64 GB of RAM, and an AMD Radeon RX 5700 XT graphics card with 8 GB of dedicated memory.

## 3. Results

### 3.1. Model Training

During the training process, our segmentation model showed a positive performance. It began with an initial loss of function of 0.6258, but quickly converged, showing a significant reduction in the loss of function with each subsequent epoch. In the 25th epoch, the loss had decreased to an impressive value of 0.0348. Simultaneously, the model’s accuracy consistently increased, reaching 98.61% in the 25th epoch. This progress was also confirmed through visual verification with the pathologists’ annotations, which demonstrated almost perfect correspondence (Figure 3). These results indicate not only the model’s capability to effectively learn from the provided dataset, but also its stability, as evidenced by the absence of any overt signs of overfitting (Figure 4).

### 3.2. Model Validation

The interobserver agreement between pathologists through Lin’s concordance correlation coefficient was optimal in eight out of ten cases and substantial in the remaining two cases, as detailed in Table 1.

According to Altman [29], the agreement between the model and pathologist B was optimal (CCC: 0.9400, IC: 0.8736, 0.9728). Similar results were obtained for the comparison between the model and pathologists D and F, whose CCCs were 0.9291 (IC: 0.8560, 0.9680) and 0.9170 (IC: 0.8813, 0.9672), respectively. Furthermore, the agreement between the model and pathologist C was substantial (CCC: 0.7918; IC: 0.6529, 0.8774), as well as agreement between the model and pathologist E (CCC: 0.7785; IC: 0.628, 0.8759). Finally, the agreement between the model and the mean of the pathologists was optimal, with CCC 0.9170 (IC: 0.8399, 0.9622). The results are summarized in Table 2.

Similarly, the agreement between our model and the five expert pathologists, verified using Bland–Altman’s plots, was also satisfactory overall. Indeed, in all comparisons, all bias lines were near 0, while all observation were within the agreement ranges (Figure 5, Figure 6, Figure 7, Figure 8 and Figure 9).

Agreement between our model and the residents was not as satisfactory. In fact, in all cases, the agreement was only substantial (Table 3): the agreement between the model and Resident 1 was 0.659; the agreement between the model and Resident 2 was 0.721; and the agreement between the model and Resident 3 was 0.615, while the agreement between the model and the mean of the residents was 0.776.

After approximately six months, two expert pathologists and a resident learned to use the proposed algorithm. Upon applying the algorithm to a set of 20 cases from the BOMs series, all three users agreed with the model’s automated assessments and decided to adopt these evaluations as the definitive. This unanimous agreement on the model’s evaluation effectively eliminated inter-observer variability. 

Regarding the opinions of the users concerning usability, the rapidity of evaluation, and the correctness of evaluation of our model, the results are summarized in Table 4.

Finally, evaluation of the performance of our model across different scanners provided superimposable results both in terms of segmentation (Figure 10) and in terms of the cellularity assessment performed by our model (Table 5).

## 4. Discussion

Our study shows that the tool we developed quantifies the cellularity of BMBs equivalently to experienced pathologists, and allows for the assessment of the performance of trainee pathologists with good user acceptability. Regarding the technical aspects of our tool, we chose to develop a fully convolutional network (FCN) that classifies every single pixel, thus allowing for precise differentiation between various types of tissues. Additionally, the use of the InceptionV3 architecture up to the “mixed7” layer provides a robust foundation for feature extraction, which has proven effective in processing histological images, particularly in the hematological field [31]. Furthermore, to ensure maximal reproducibility of our study, we trained our model using WSIs obtained from a slide scanner system (Ventana DP 200) approved for in vitro diagnostic use [4].

Unfortunately, despite high concordance among experienced hematopathologists, our results showed that inter-observer variability was evident, with a wide range in Lin’s coefficient. This may in part be explained by the fact that hematopathologists typically express assessments of bone marrow cellularity in increments of 5% (e.g., 10%, 15%, 20%, etc.), while the AI system provides a precise numerical value to the second decimal place. Consequently, the concordance between hematopathologists, as well as between hematopathologists and the model, could be influenced by this specific procedure of evaluation performed by the hematopathologist, which represents a more generalized and “rounded” assessment value.

Regarding the agreement between the AI-based tool and the human eye, our results were optimal. The models demonstrated high concordance with the assessments of five expert hematopathologists. Additionally, the concordance between the AI model and the average assessments of the five hematopathologists was also optimal. Furthermore, evaluations using Bland–Altman analysis were substantially positive across all comparisons, with a bias line near zero and most measurements falling within the limits of concordance. In addition, the lower agreement observed between the model and the residents indicates that, for unexperienced pathologists, cellularity assessment can be less reproducible.

Despite the involvement of a higher number of observers in our study (five expert pathologists and three residents), our results are consistent with the existing literature. In their respective studies, Nielsen et al. [23] and van Eekelen et al. [24] developed automated tools for assessing cellularity in BMBs, achieving optimal concordance with the human observations. However, their tests were limited to single observer, whereas our model’s performance was compared with that of five expert hematopathologists. Sarkis et al. [32] also compared the performance of a model for BMB cellularity assessment with that of four pathologists, obtaining a high level of concordance with all four. However, the model developed by Sarkis et al. required semi-automated interaction, necessitating user involvement for performing a white color balance, making three annotations (for the tissue to be evaluated), and selecting the background and artifacts before analyzing the entire BMB. 

A strength of our work is that, following training and a lag time of 6 months, when we asked three users to apply the model to 20 randomly selected BMBS, all concurred with the assessments provided by the model and were willing to adopt the assessments as definitive. This unanimous agreement to adopt the model removes the inter-observer variability of human assessments. 

Evaluation of the model’s usability (usability (learning difficulties and autonomy in using the model) indicated that, while users found the rapidity and accuracy of the model to be favorable, they were not satisfied with its general usability. These findings align with those from previous studies evaluating pathologist-reviewed models. In Dy et al.’s work [33], which assessed a tool for KI 67 evaluation in breast cancer, and Steiner et al.’s study [34], which evaluated a tool for prostate biopsy assessments, users consistently praised the models’ accuracy and supported their integration into clinical practice. However, neither study addressed the models’ usability or the learning curves associated with their use. It is important to note that both studies developed user interfaces that allowed pathologists to easily select cases for examination.

Since previous studies have demonstrated how the optical and computed properties of WSIs are affected by technical differences in slide scanners, potentially influencing the performance of AI-based tools [9], we found it necessary to evaluate our model’s performance using WSIs obtained from different scanners. Duenweng et al. [10] demonstrated that the optical and computed properties of WSI from different scanners can affect AI-based tools, particularly when applied to low-resolution images (i.e., low magnification). In our study, we worked (for both training and validation of the model) on patches obtained at an intermediate magnification (20×), which ensured that the performance of our model was not adversely affected by the differing properties of the two scanners. Therefore, unlike these previous data, where model performance was influenced by the scanner type, our data confirm that the model’s performance was remarkably consistent, in terms of both the method of segmentation and the cellularity assessment. 

A major limitation of our model was usability by clinicians, particularly those with little computer experience, and particularly when under time pressure. It would benefit from a user-friendly interface that requires the pathologist to simply select the case to be analyzed. Moreover, our model’s validation was conducted in a single center, which may limit the generalizability of our findings. For real use in clinical practice, our model should be tested in a multicenter study in order to test its reproducibility and its efficacy on a larger number of cases under different conditions of use (hardware, network, clinical routine, etc.). The sole assessment of cellularity is also limiting. The model’s utility could be enhanced if other important features of Ph-MPNs could be added, such as individuation of the proliferation of one or more cellular lines, evaluation of increasing numbers, and morphological details of the megakaryocytes and of a shift in maturation and BM fibrosis. However, automated assessment of these features is more complex than cellularity evaluation; therefore, future studies would require the development of more complex neural networks capable of carrying out this type of evaluation. 

## 5. Conclusions

In conclusion, our automated system for assessing cellularity in BMBs compared favorably with the evaluations by five experienced pathologists using an optical microscope, demonstrating optimal results and the ability to eliminate interobserver variability in the evaluation of cellularity from BMBs of Ph-MPNs. The low variability, accuracy, and speed of our model potentially make it a useful adjunct to expert histopathologist assessments, reducing the time needed for quantitative assessments and acting as a learning resource for pathology trainees.

## Figures and Tables

**Figure 1 cancers-16-01687-f001:**
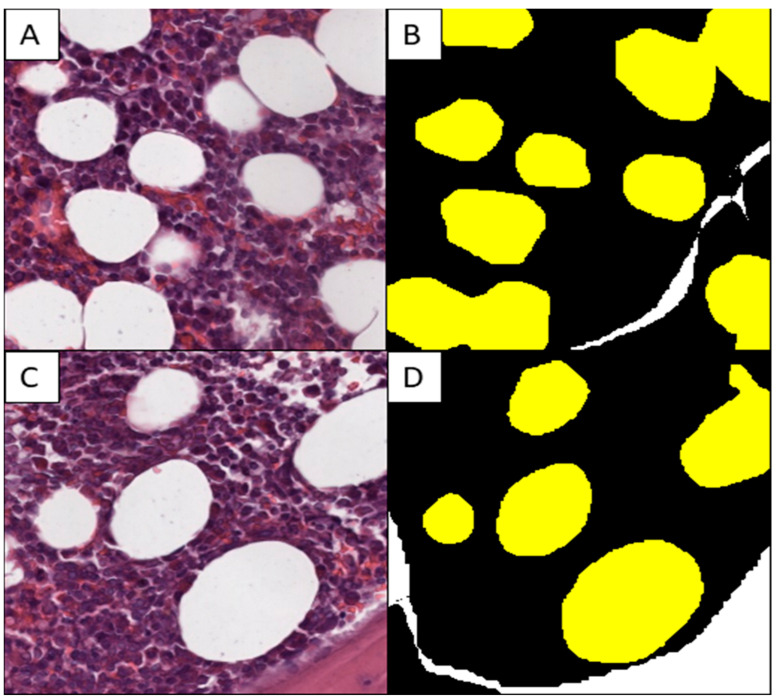
Example of a training set image with tiles ((**A**–**C**); H&E, 20×) and corresponding labels (**B**–**D**).

**Figure 2 cancers-16-01687-f002:**
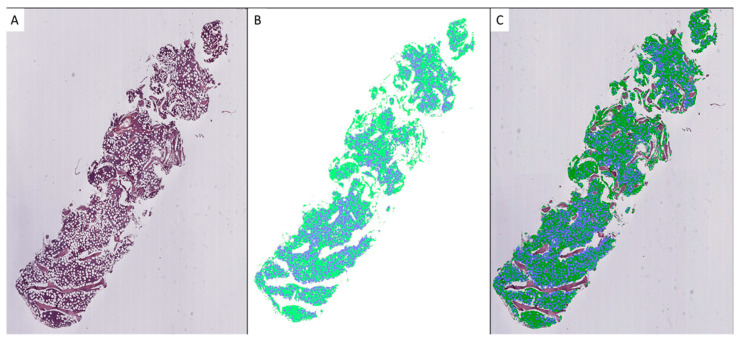
Workflow of segmentation process: WSI (**A**), segmentation result (**B**), WSI with overlaid segmentation image (**C**). Green indicates hematopoietic tissue and blue indicates adipose tissue.

**Figure 3 cancers-16-01687-f003:**
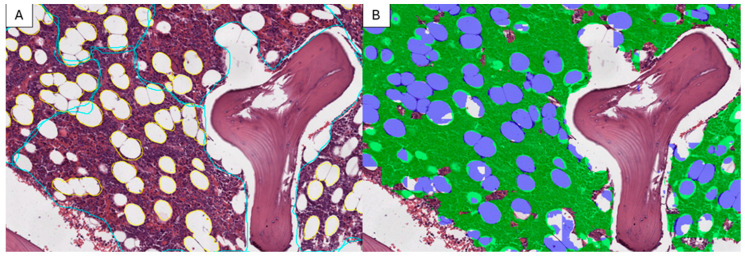
Comparison between pathologists’ annotations ((**A**) cyan: hematopoietic tissue; yellow: adipose tissue) and the model’s result ((**B**) green: hematopoietic tissue; blue: adipose tissue).

**Figure 4 cancers-16-01687-f004:**
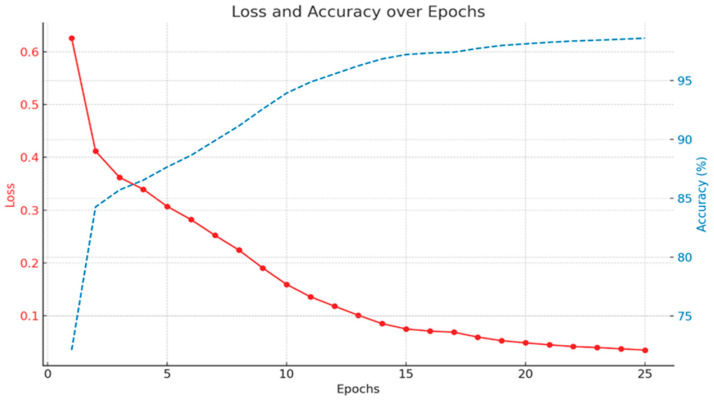
Graph of the training process: loss of function (red) and accuracy (blue).

**Figure 5 cancers-16-01687-f005:**
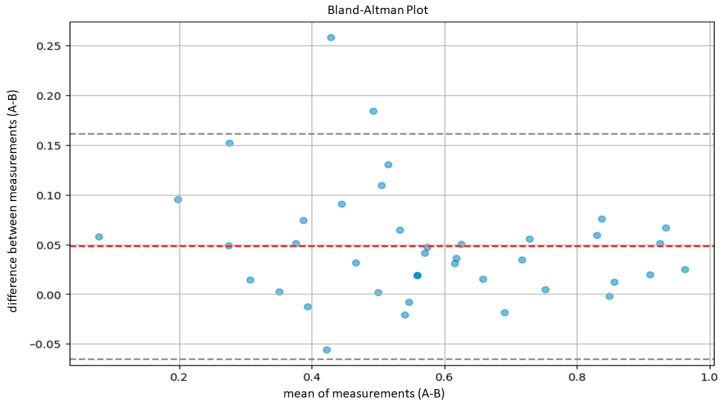
Bland–Altman plot to measure agreement between the model (A) and pathologist B.

**Figure 6 cancers-16-01687-f006:**
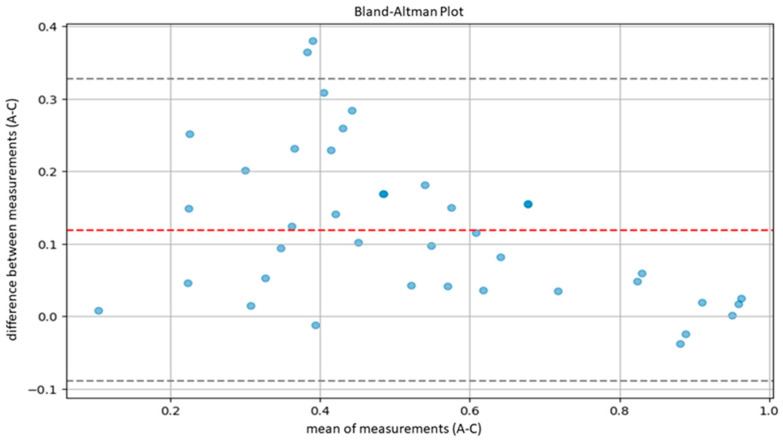
Bland–Altman plot to measure agreement between the model (A) and pathologist C.

**Figure 7 cancers-16-01687-f007:**
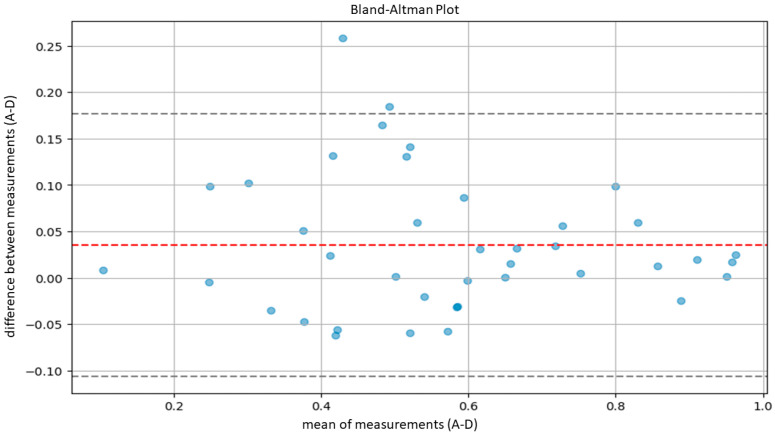
Bland–Altman plot to measure agreement between the model (A) and pathologist D.

**Figure 8 cancers-16-01687-f008:**
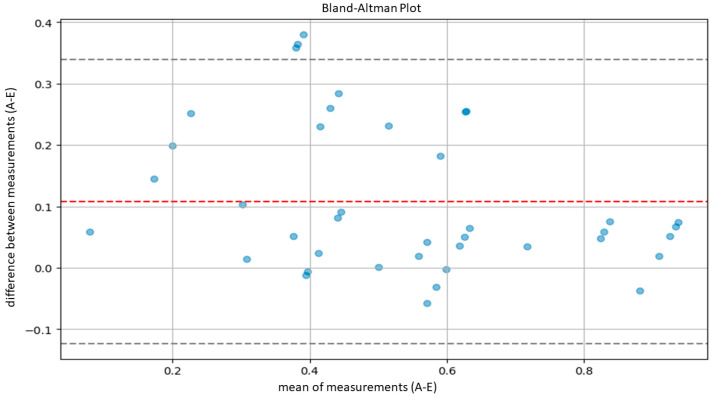
Bland–Altman plot to measure agreement between the model (A) and pathologist E.

**Figure 9 cancers-16-01687-f009:**
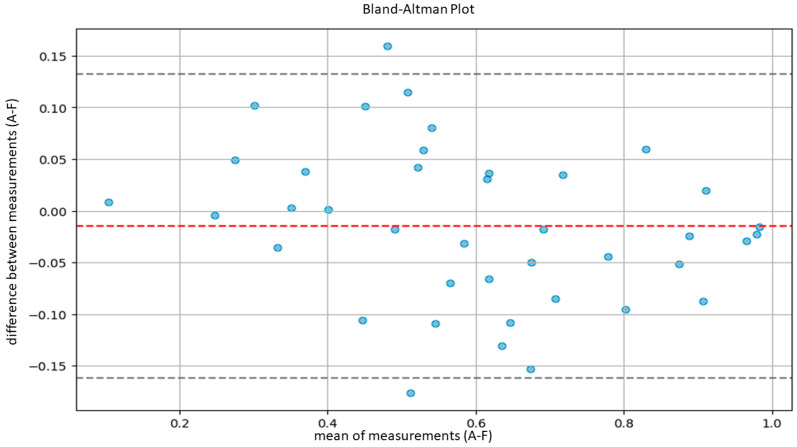
Bland–Altman plot to measure agreement between the model (A) and pathologist F.

**Figure 10 cancers-16-01687-f010:**
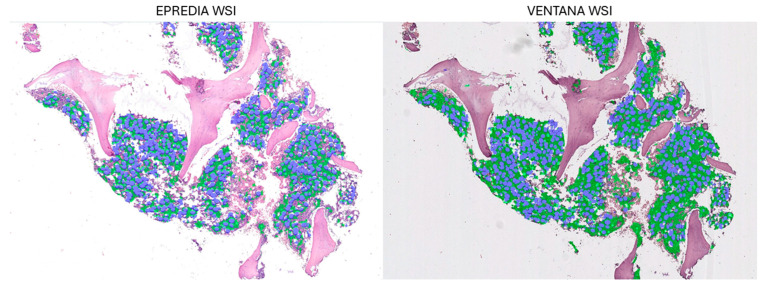
Details of a selected WSI for rescanning with the Epredia P1000. A comparison between the segmentation results of the WSI processed by Epredia and those processed by Ventana.

**Table 1 cancers-16-01687-t001:** Agreement between pathologists (B-F).

Couple of Pathologist	Lin’s Concordance Correlation Coefficient	Confidence Interval
B-C	0.887	0.789, 0.938
B-D	0.967	0.941, 0.982
B-E	0.886	0.788, 0.944
B-F	0.899	0.813, 0.944
C-D	0.877	0.784, 0.925
C-E	0.951	0.906, 0.973
C-F	0.773	0.663, 0.860
D-E	0.878	0.794, 0.929
D-F	0.902	0.715, 0.952
E-F	0.784	0.648, 0.873

**Table 2 cancers-16-01687-t002:** Agreement between model (A), each pathologist (B-F), and the mean of pathologists (G).

	Lin’s Concordance Correlation Coefficient (CCC)	Confidence Interval	
A-B	0.9400	0.8736	0.9728
A-C	0.7918	0.6529	0.8774
A-D	0.9291	0.8560	0.9680
A-E	0.7785	0.6288	0.8759
A-F	0.9386	0.8813	0.9672
A-G	0.9170	0.8399	0.9622

**Table 3 cancers-16-01687-t003:** Agreement between model, each resident, and mean of residents.

	Lin’s Concordance Correlation Coefficient (CCC)	Confidence Interval	
Model–Resident 1	0.659	0.508	0.771
Model–Resident 2	0.721	0.546	0.845
Model–Resident 3	0.615	0.432	0.755
Model–mean of residents	0.664	0.513	0.776

**Table 4 cancers-16-01687-t004:** Opinion of the users about the model (1: completely insufficient; 2: acceptable; 3: sufficient; 4: good; 5: optimal).

	Usability	Rapidity	Correctness
User1	2	5	5
User2	2	5	4
User3	3	4	5

**Table 5 cancers-16-01687-t005:** Cellularity assessment performed by the model on Epredia WSI and on Ventana WSI.

	Epredia WSI	Ventana WSI
Slide 1	50.85%	50.15%
Slide 2	64.98%	65.05%
Slide 3	99.35%	97.47%
Slide 4	58.62%	59.14%
Slide 5	84.77%	86.26%

## Data Availability

All codes and data are available upon reasonable request by email to the corresponding author.

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
