# Peer review of "Development of an Artificial-Intelligence-Based Tool for Automated Assessment of Cellularity in Bone Marrow Biopsies in Ph-Negative Myeloproliferative Neoplasms"

_cancers, 2024, doi:10.3390/cancers16091687_

Round 1

Reviewer 1 Report

Comments and Suggestions for Authors

I congratulate the authors on their work and I believe that developing these kind of algorithms will be of great help to pathologists. However I do have some points to raise, please find them below.

1. The aim of this study is to develop an accurate AI-based tool for quantification of BM cellularity, to improve uniformity and repeatability (reproducibility?) of cellularity evaluation beyond the expertise of the individual pathologist. However, as I understand, assessment of cellularity is only one subtask of many others (proliferation of one or more cellines, the number of morphological details of megakaryocytes and a left or right shift in maturation and BM fibrosis). Why do you think solving the one subtask (that seems maybe least subjective of all above mentioned tasks) would result in overall improved uniformity and reproducibility? 

You may want to consider to focus on workflow improvements as well.
Because if one thing, at least it will be a handy tool for pathologists, making their work easier. For pathology laboratories that have to implement these algoritms, they need other incentives as well, like workflow improvements (time- and/or cost savings, as algorithms themselves are not for free). So I would suggest to incoporate this as well.

In addition, I miss the part in the Discussion on how this should be used or implemented in clinical practice (AI-assisted, screening, independent AI?). Can the author elaborate on this? 

2. The used biopsies are all from the same archive (single pathology department) and scanned on the same scanner. This could have implications for generalizabilty and usibility of the algorithm outside this specific clinic. Could you elaborate some more on this in the discussion?

3. Line 90 states that 15 cases were used for the training set, but then line 94 says 7 of those 15 were randomly selected. Why is that and what happened to the other 8? This may be confusing to the reader.
I am not a statistical expert, but 7 cases does not seem a lot of samples to train a dataset. Can the authors elaborate on this a bit more?

4. The part on the statistical analysis is quite short and does not explain how agreement is measured (and also not that you used definitions of Altman to define agreement). I assume cellularity is measured as pecentages, so what is agreement? Exactly the same % or is it measured in ranges? (I only read in the Discussion that it is measured by pathologists in range of 5%, this clearly should be mentioned in the Methods section. From the Discussion I understand that agreement is exactly the same %? Which the authors state that could influence concordance). This is a trivial point in the study design (if it is measured in ranges), which is currently not addressed.

Also why did the authors choose to do pairwise measurements of those 5 pathologists? 

Lastly on the Methods section, can the authors also explain how they determined the agreement ranges in the Bland-Altman plots? From reading the results questions I see what measures they used, but some of the above mentioned details are not explained in the methods section.

6. The first two paragraphs of the Discussion would be more suitable for the introduction part. These two paragraphs highlight the essence of this study in a very concise manner.

7. I believe there is a typo in the abstract. 55 BMBs were selected and than it says 50 were randomly selected and the remaining 40 were used for model verification and performance. I think the 50 should be 15 (as correctly mentioned in the methods section).

In general I see the potential of this paper (and all the work that went into it), but I feel the paper could be structured better to improve readibility. I also think the paper would be a lot stronger when also validated outside this single clinic, and when the authors elaborate on how this should be used in daily practice. 

Comments on the Quality of English Language

Sometimes signaling wors like 'indeed' and 'thus' are used incorrectly which can be a bit misleading and confusing. For example line 175: similary and also (use one but not both), use of "Indeed" in line 199, "Thus" in line 202. These words do not make sense in these sentences.

Reviewer 2 Report

Comments and Suggestions for Authors

In this article authors developed an AI based tool for automated assessment of cellularity in bone marrow biopsies in Ph-negative myeloproliferative neoplasms. Automated tools are highly required to avoid inter and intra-observer variability. QuPath is a very powerful software for digital scanning. Authors used 15 BMBs for training purpose and 40 remaining BMBs for validation. Authors also got BMBs cellularity evaluated by 5 expert pathologists.  It is impressive to achieve 98.61% accuracy by 25th epoch. This manuscript is well written. However, few typo errors were found. One of them is in abstract instead of 15, fifty was typed for training set. 

Reviewer 3 Report

Comments and Suggestions for Authors

This is a study of cell density assessment of bone marrow biopsies in patients with Ph-negative myeloproliferative neoplasms using AI technology. Although there is some novelty in using AI-based technology, some concerns exist.

#1. A more detailed description of the findings in Figure 1 is needed: the yellow colored areas in Figure 1 are considered to be adipocytes, but what do the white linear colored areas mean?

#2. A more detailed description of the findings in Figure 2 is needed, e.g., what each color means.

#3. The authors state, "The main histological findings in this setting include overall cellularity, the proliferation of one or more cellular lines, the number and morphological details of the megakaryocytes, a left or right shift in maturation and BM fibrosis." However, whether this AI-based model can be used for such purposes is questionable. The method is based on segmentation as a region rather than on individual cells. For example, it seems possible to determine overall cellularity and BM fibrosis but not the proliferation of one or more cellular lines, the number and morphological details of the megakaryocytes, and a left or right shift in maturation. I would like to see a separate analysis of the usefulness of this AI model for each of the above factors.

#4. Comparative data between the authors' annotations and the AI-based model, using highly magnified images, would be appreciated.

#5. Expert pathologists: Please provide the number of years of career as a hematopathologist for each of the five expert pathologists, even if it is only roughly.

#6. To ensure that "The diagnosis may be challenging, being not reproducible and dependent on the pathologist's experience." in addition to the expert pathologist, the results of the diagnosis in younger doctors, such as residents, should be presented.

#7. There appear to be some limitations, including the above point, which are not adequately discussed.

#8. As the authors are using patient specimens, an IRB review is required.

Comments on the Quality of English Language

Minor editing of English language required.

Round 2

Reviewer 1 Report

Comments and Suggestions for Authors

I can see the tremendous work the authors put into their revision and I would like to thank them for that. However, I think there are still some remaining issues, before this paper can be accepted.

1. The discussion is extremely short and is not comprehensive (the authors just moved 2 paragraphs to the introduction). The discussion really needs to be improved by putting the research findings into context (citing other papers investigating this), elaborating on the future steps for research/implementation (will it be used as AI-assistance? affordablility?). The last two paragraphs of the conclusion should for example be in the discussion. But also the future steps of the authors with this algorithm, what is next, validation on another scanner in an external cohort?
A particularly helpful tool to write a comprehensive discussion that I often use myself is this one: Question-Tool-Discussion-.pdf (wetenschappelijkschrijven.nl)

2. My point on the scanner is not addressed by the authors. They state that they used an approved scanner, but so does everybody in diagnostic pathology. The point is that you train an algorithm on one scanner only, this may have consequences for the generalizability of your algorithm if people with other scanners were to use it. This is the point I would like to see addressed.

3. The authors first assessed inter-pathologist variability, which is informative, but it is known that there is interobserver variability. Then they compare the algorithm with individual pathologists, which in fact is again a measure of agreement of pathologist with a second read (this time by AI). The outcome to me is not that informative. What you would actually want to know is whether an AI-assisted pathologist (because that is how you would use in daily practice I assume) would show less inter-observer variability (compared to a reference standard) to draw the conclusion that AI will reduce interobserer variability. Then the claim that this algorithm may reduce interobserver variability is justified. Otherwise the only real conclusion that can be drawn from this study is that the algorithm agrees with expert hematopathologists at similar levels as two hematopathologists among eachtother and worse agreement with residents (which makes sense, they are still learning). Therefore, the algorithm is as good as an experiences hematopathologist. 

Comments on the Quality of English Language

English language and signalling words can still be improved, especially the recently added sentences in the introduction. If possible please let it be revised by a native speaker to improve readability, this would really help to get your message to readers. 

Round 3

Reviewer 1 Report

Comments and Suggestions for Authors

I would like to thank the reviewers for their extensive work, especially on the Discussion.

I have the following remaining points of concern:

1. Please have the text edited/read by a native English speaker. Signalling words as 'as soon as' in line 102 are used incorrect and are confusing. I assume the authors mean: 'as fast as'. Another example is line 108: 'Previous studies have been made', which should be "previous studies have been conducted" or 'previous studies have made claims/are claiming' etc.
This will enhance readability very much and keep readers interested in reading. Now I sometimes struggle to get what the authors mean, and a reader will lose interest.

2. The first part of the Discussion is a lot better, but from line 423 it gets confusing again and this part is not in line with the rest of the Discussion. It is also a repetition of the Methods section (in part). Again, compliment for the extensive revision of the Discussion, but please structure it a bit more to improve readibility, because you will lose the reader here. Also line 436 again compares the current study to previous literature, which should be one combined paragraph with the previous paragraph on this in the Discussion.

3. I like the extra part on 20 cases and opinion of the users, but what do they mean by usability, what did they ask the users? Can they show their questions?

4. I do not understand the claim that the algorithm fully eliminates inter-observer variability. I don't think that can be claimed when the AI agrees with an expert pathologist in a few cases. A better conclusion would be that it reduces inter-observer variability and reaches full agreement in ...% of cases.

In general, the paper is already much improved, but readability remains an issue for me. I have read this paper 3 times now and I keep struggeling and figuring out what the authors mean or which direction paragraphs are going. A review of a native English speaker and a better structure of mainly the Discussion need to be performed to make this manuscript acceptable. 

Comments on the Quality of English Language

See comments above.

Author Response

Thanks to the reviewer for these additional comments that will make the manuscript even clearer and improve its readability.

In replying point-by-point to the comments, the remarks of the Reviewer1 are reported in black, the Authors’ comments in blue, and the added text (for third round) is reported in green in the revised manuscript. In our response, we also indicated page and line for all changes within the manuscript.

Firstly, we proceeded with a new general revision of the English improving the readability of the manuscript.

Regarding the reviewer's comments:

  1. Please have the text edited/read by a native English speaker. Signalling words as 'as soon as' in line 102 are used incorrect and are confusing. I assume the authors mean: 'as fast as'. Another example is line 108: 'Previous studies have been made', which should be "previous studies have been conducted" or 'previous studies have made claims/are claiming' etc.

This will enhance readability very much and keep readers interested in reading. Now I sometimes struggle to get what the authors mean, and a reader will lose interest.

  1. The first part of the Discussion is a lot better, but from line 423 it gets confusing again and this part is not in line with the rest of the Discussion. It is also a repetition of the Methods section (in part). Again, compliment for the extensive revision of the Discussion, but please structure it a bit more to improve readibility, because you will lose the reader here. Also line 436 again compares the current study to previous literature, which should be one combined paragraph with the previous paragraph on this in the Discussion.

A: We have further revised the English, particularly by reworking all the paragraphs of the article with the assistance of a native-English speaker. Additionally, we have restructured the discussions in a paragraph for discussing all the results and another for future developments of the work. We hope that this will enhance the readability of the article.

3 I like the extra part on 20 cases and opinion of the users, but what do they mean by usability, what did they ask the users? Can they show their questions?

A: We explained to the users that by 'usability,' we were referring to whether there were any learning challenges associated with using the model and whether participants felt satisfied with their independence in using it. We clarify this concept in method section (page 5, line 172-173)

4: I do not understand the claim that the algorithm fully eliminates inter-observer variability. I don't think that can be claimed when the AI agrees with an expert pathologist in a few cases. A better conclusion would be that it reduces inter-observer variability and reaches full agreement in ...% of cases.

We thanks’ the reviewer1 for their comment, which allow us to clarify a point that only partially we address in our manuscript: Our model autonomously identifies adipose and hematopoietic tissue using two Python scripts, providing an accurate cellularity’s assessment result. When users operated the model independently, they fully agreed with the assessments automatically generated by the model, adopting them as the definitive evaluations of cellularity. Consequently, our model, which provided the same evaluation to all three users, effectively eliminated inter-observer variability. We have clarified this point in the results section (page 11, lines 314-317) and in the discussion section (page 13, lines 389-391).
